# Topical Administration of a Novel Acetylated Tetrapeptide Suppresses Vascular Permeability and Immune Responses and Alleviates Atopic Dermatitis in a Murine Model

**DOI:** 10.3390/ijms232113498

**Published:** 2022-11-04

**Authors:** Bokyung Sung, Yi-Yong Baek, Young-Myeong Kim, Ji Chang You

**Affiliations:** 1Avixgen Inc., 2477, Nambusunhwan-ro, Seocho-gu, Seoul 06725, Korea; 2Department of Molecular and Cellular Biochemistry, School of Medicine, Kangwon National University, Chuncheon 24341, Korea; 3National Research Laboratory for Molecular Virology, Department of Pathology, School of Medicine, The Catholic University of Korea, Seoul 06591, Korea

**Keywords:** Ac-RLYE, atopic dermatitis, inflammation, vascular permeability, cytokines

## Abstract

Although the pathogenesis of atopic dermatitis (AD) remains to be fully deciphered, skin barrier abnormality and immune dysregulation are known to be involved. Recently, the vascular endothelial growth factor (VEGF)/VEGF receptor (VEGFR) system has also been implicated in the pathogenesis of this multifactorial chronic inflammatory skin disorder. Previously, we showed that a novel tetrapeptide, *N*-acetyl-Arg-Leu-Tyr-Glu (Ac-RLYE), inhibits angiogenesis and vascular permeability effectively by selectively antagonizing VEGFR-2. The current study aimed to investigate the pharmacological effect of Ac-RLYE on AD in vitro and in vivo. The in vitro experiments demonstrated that Ac-RLYE inhibited VEGF-induced vascular permeability in endothelial cells. Moreover, in an in vivo animal model of AD, Ac-RLYE relieved AD-like symptoms such as ear thickness and dermatitis severity scores and infiltration of immune cells, including mast cells and eosinophils. Ac-RLYE inhibited IgE secretion, restored the skin barrier protein filaggrin level, and markedly downregulated gene expression of AD-related Th1, Th2, and Th17 cytokines. Collectively, these findings suggest that Ac-RLYE would be useful for the treatment of AD and associated inflammatory skin disorders.

## 1. Introduction

Atopic dermatitis (AD) is characterized by an aberrant immune response and skin barrier disruption. AD is a recurrent, chronic inflammatory skin disease affecting ~20% of children and 2.1–4.9% adults worldwide [1,2]. There has been no cure yet, and conventional drugs are used to relieve temporary itching and dermatitis symptoms. In general, topical or systemic corticosteroids are used, but there have been serious side effects such as skin thinning or secondary infection due to expanded blood vessels. The use of calcineurin inhibitors, tacrolimus or pimecrolimus, also causes adverse effects such as itching and burning. Biological drugs, such as antibodies to interleukin (IL)-4 or IL-13, have been approved for moderate to severe AD patients, but there may be limitations such as resistance development and high drug costs. Hence, it is necessary to develop a novel AD therapy based on the understanding of the pathophysiology of AD to minimize side effects and provide a complete cure [3].

Vascular endothelial growth factor (VEGF) is a well-known, potent, angiogenic, and vascular permeability factor [4]. The VEGF family of proteins activates three types of VEGF receptors (VEGFRs). Among them, the VEGF-A/VEGFR-2 axis elicits the most potent angiogenic activity and vascular permeability. VEGF has previously been shown to be a contributor to skin inflammation in AD [5]. This is supported by the evidence that VEGF causes hyperpermeability of vessels leading to edema and spongiosis [6]. Furthermore, VEGF induces chemotaxis of myeloid cells and activates endothelial cells, leading to the increased expression of adhesion molecules in these cells and subsequent promotion of tissue infiltration of leukocytes [7,8]. Moreover, increased VEGF levels have been observed in skin lesions and plasma of AD patients [5]. Thus, inhibition of VEGF would likely be useful as a potential treatment for AD.

To overcome these unmet needs, such as novel therapeutics based on the etiology of AD, we developed an acetylated tetrapeptide (*N*-acetyl-Arg-Leu-Tyr-Glu, hereafter Ac-RLYE) that selectively targets VEGFR-2 signaling, which is closely associated with increased vascular permeability. The acetylated tetrapeptide, Ac-RLYE, blocks the growth and metastasis of various cancer cells by suppressing angiogenesis in vivo [9,10,11]. Recent reports have also shown that Ac-RLYE has therapeutic potential against diseases caused by retinal neovascularization, such as diabetic retinopathy and age-related macular degeneration [12,13].

Most AD therapies based on controlling the immune response, such as topical corticosteroids or immunosuppressants, focus on inhibiting the immune response elicited by already-produced immune substances and their movement to skin lesions. Thus, there is currently no approved anti-angiogenic-based therapy for AD treatment. Therefore, we have made such attempts to reduce vascular hyperpermeability by restoring the integrity of vascular vessels, thereby inhibiting plasma leakage and the movement of inflammatory cells into atopic skin tissue lesions. This ensures a differentiated mechanism of action that can block excessive skin inflammatory reactions and provide a fundamental treatment for atopic patients at all levels.

In the present study, we determined whether Ac-RLYE, a novel VEGFR-2-specific antagonist, ameliorates AD-like symptoms by reducing vascular permeability in a murine AD model. Our results showed that Ac-RLYE reduced vascular permeability through the inhibition of vascular barrier disruption, thereby inhibiting the infiltration of inflammatory cells into AD lesions and alleviating AD symptoms.

## 2. Results

### 2.1. Ac-RLYE Blocks Vascular Endothelial Growth Factor-Induced Permeability in HUVECs

Vascular endothelial growth factor (VEGF) is a cytokine that promotes the survival, migration, and proliferation of endothelial cells. High levels of VEGF and VEGFR are found in the plasma and AD lesions of AD patients [14,15,16,17]. Thus, highly activated VEGF–VEGFR signaling in AD can lead to vessel dilation, vascular hyperpermeability, and continuous extravasation of inflammatory cells and fluid into the inflamed tissue [16]. Thus, we investigated whether Ac-RLYE has a protective effect on vascular endothelial cells against VEGF-induced permeability by preserving endothelial barrier integrity. To determine the effect of Ac-RLYE on endothelial barrier integrity, we examined the effect of Ac-RLYE on the stability of the tight junction (TJ) protein ZO-1 and the adherens junction (AJ) protein VE-cadherin by immunofluorescence in HUVECs. In confluent HUVECs under normal conditions, ZO-1 and VE-cadherin were predominantly found at cell–cell contact sites, which confirmed junction formation. Treatment with VEGF disrupted the ZO-1 and VE-cadherin junction compared to the untreated control (Figure 1A). Ac-RLYE significantly attenuated this VEGF-induced cellular junction disruption (Figure 1A,B). Together, these results suggest that Ac-RLYE inhibits VEGF-induced vascular hyperpermeability by blocking junction disruption.

### 2.2. Ac-RLYE Alleviated AD Symptoms in HDM-Challenged NC/Nga Mice

To investigate whether Ac-RYLE has the therapeutic potential to alleviate AD, we induced AD in mice by first applying HDM to the ears and dorsal area of NC/Nga mice followed by Ac-RLYE to the ears and dorsal area once daily for 4 weeks (Figure 2A). Tacrolimus 0.1% ointment (Protopic^®^), a macrolide calcineurin inhibitor approved for use in patients with AD, was used as a positive control. The body weight changes were not significantly different between vehicle- and Ac-RLYE-treated groups (Figure 2B). We found that the ear thickness increased after treatment with HDM (vehicle-treated) during the experimental period, and this increase was significantly attenuated by Ac-RLYE in a dose-dependent manner (Figure 2C, left). Interestingly, the effect of Ac-RLYE on ear thickness was slightly better than that of tacrolimus at doses above 0.02% (Figure 2C, right).

We also performed a macroscopic analysis of skin lesions to evaluate the severity of the dermatitis. Representative photographs for each group of animals are shown in Figure 2D. On Day 48, the dorsal skin of mice treated with HDM showed severe AD skin symptoms with prominent erythema/hemorrhage, edema, excoriation/erosion, and scaling/dryness. The application of the highest dose of Ac-RLYE (0.1%) in mice led to a noticeable reduction in the dermatitis scores, and the effect was equivalent to that observed in the tacrolimus-treated group (Figure 2D). The scores for each group were as follows: 7.8 ± 0.8 (vehicle control), 7.4 ± 0.8 (0.004% Ac-RLYE), 5.8 ± 2.3 (0.02% Ac-RLYE), 4.5 ± 2.4 (0.1% Ac-RLYE), and 4.5 ± 2.4 (tacrolimus).

Atopic dermatitis-induced skin lesions are characterized by hypertrophy, hyperkeratosis, acanthosis, parakeratosis, and massive infiltration of inflammatory cells. Next, we performed histological analysis to further evaluate the anti-inflammatory effects of Ac-RLYE. The results revealed hypertrophy and hyperkeratosis in the dorsal epidermis of HDM-treated mice (Figure 3A,B). Compared with the vehicle-treated HDM group, the scores of both hypertrophy and hyperkeratosis in epidermis decreased in Ac-RLYE-treated groups (Figure 3B). We also observed that the all Ac-RLYE-treated groups showed less infiltration of inflammatory cells in both epidermis and dermis compared to the vehicle-treated HDM group (Figure 3C). Collectively, these results clearly demonstrated the therapeutic potential of Ac-RLYE in the treatment of AD-like lesions in the HDM-induced murine model.

### 2.3. Ac-RLYE Lowered the Levels of White Blood Cells (WBCs) in the Blood of HDM-Challenged NC/Nga Mice

WBCs, including subtypes such as neutrophils, lymphocytes, monocytes, eosinophils, and basophils, are activated in the blood of patients with AD [18,19]. In particular, activation of eosinophils is found in the majority of AD patients, which is closely correlated with disease activity [20]. To evaluate whether topical application of Ac-RLYE can decrease the levels of inflammatory cells induced by repeated HDM challenge, we assessed inflammation in the blood of mice using differential counting at Day 48. Compared to the vehicle-treated control, treatment with Ac-RLYE dose-dependently lowered the number of leukocytes, neutrophils, and eosinophils, indicating that Ac-RLYE suppresses inflammation by decreasing the number of WBCs in the blood (Figure 4).

### 2.4. Ac-RLYE Reduced Mast Cell Infiltration and Plasma IgE

Since increased numbers of mast cells have been reported in the skin lesions of patients with AD or experimental AD murine models [21,22], we investigated the effect of Ac-RLYE on the local infiltration of mast cells in the AD-like skin lesion of mice. Toluidine blue staining was performed for mast cell visualization. Compared to the vehicle-treated group, the accumulation of mast cells (red arrows in Figure 5A) was reduced in the Ac-RLYE-treated group (Figure 5A).

As elevated plasma levels of IgE are a critical indicator of AD development and are involve in mast cell activation [23], we examined IgE levels in AD mice at Day 48. As expected, the topical application of Ac-RLYE significantly lowered plasma IgE compared to the vehicle-treated group (Figure 5B).

### 2.5. Ac-RLYE Restored Skin Barrier Function-Related Protein Level

The loss of skin barrier function is one of the characteristics of AD [23]. Hence, we performed an ELISA to measure the protein level of filaggrin (FLG), a structural protein that plays a crucial role in skin barrier functions, in the dorsal skin of mice. The repeated application of HDM led to a noticeable reduction in FLG levels; the FLG level of the vehicle-treated AD group was 1150.3 ± 146.8 pg/mL. The protein level of FLG was significantly restored in the Ac-RLYE groups in a dose-dependent manner: the 0.1% Ac-RLYE group showed the highest FLG level (1469.4 ± 105.2 pg/mL) with a similar effect to that of tacrolimus (1444.8 ± 110.5 pg/mL) (Figure 5C). These results suggest that the mechanism of action of Ac-RLYE in AD suppression may be the reduction of IgE level increases by allergen permeation, a consequence of skin barrier disruption.

### 2.6. Ac-RLYE Suppressed HDM-Induced Th1 and Th2 Cytokine Expression

An imbalance of type 1 helper T (Th) 1 and Th2, skewed toward Th2, is the underlying mechanism of AD development [24]. The Th17 cytokine has also been reported to be increased in AD, although its role is still controversial and is not as important as in psoriasis [25]. To investigate whether Ac-RLYE could attenuate the immune response, we measured the mRNA expression of cytokines secreted by Th1 *(IL-23, TNF-α*, and *IFN-γ),* Th2 (*IL-4, IL-13,* and *IL-31*), and Th17 (*IL-17A*) in the mouse dorsal tissue by qRT-PCR analysis. We observed that topical Ac-RLYE treatment remarkably inhibited the mRNA levels of Th2-type cytokines, including *IL-4, IL-13*, and *IL-31*, as well as of the Th2 chemokine *CCL17* (known as thymus- and activation-regulated chemokine (TARC)) in AD-like skin lesions. Topical Ac-RLYE treatment also inhibited the expression of Th1 cytokines, *IL-23* and *TNF-α*, but increased the expression of *IFN-γ* compared to those in the vehicle-treated group (Figure 6). In addition, the expression of *IL-17A* (often known as *IL-17*), a Th17 cytokine, was also suppressed by Ac-RLYE treatment. Intriguingly, we observed that the inhibitory effect of Ac-RLYE on the expression of *IL-31, IL-17A,* and *IL-23* was more prominent at 0.1%, the highest dose used in the current study, being similar to or even superior to that of the positive control tacrolimus. Collectively, the results demonstrate that the production of Th1 and Th2 cytokines in HDM-induced mice is downregulated by Ac-RLYE application, resulting in the suppression of AD development.

## 3. Discussion

A growing body of evidence indicates the intimate association between angiogenesis and chronic inflammatory skin diseases. However, there is an unmet need for an anti-vascular therapy for such skin diseases. Recently, the Ac-RLYE, a specific VEGFR-2 antagonist, has been reported as a novel therapeutic molecule for neovascular retinal diseases and tumors in which angiogenesis is the etiology [9,12,13]. Thus, we aimed to investigate the therapeutic potential of Ac-RLYE on AD. Using a murine model of AD, we found that topical Ac-RLYE effectively reduced AD-like symptoms and inflammatory responses. In particular, the major pathogenic factors of AD—leukocyte infiltration, skin barrier abnormality, and immune system dysregulation—were remarkably ameliorated after treatment with Ac-RLYE. It was also confirmed in vitro using endothelial cells that Ac-RLYE can suppress vascular hyperpermeability, which is one of the factors that promotes AD development. Our findings indicate that therapeutic intervention at the vasculature level, such as the regulation of vascular permeability, can be effective in the treatment of AD, and that the VEGFR-2-specific antagonist Ac-RLYE might be a promising and safe candidate for treating AD.

Accumulated evidence suggests a strong interplay between angiogenesis and inflammation in AD. Additionally, skin changes such as epidermal hyperkeratosis, acanthosis, and papillomatosis are characteristics of AD in both subacute and chronic stages, and all these AD skin changes require angiogenesis [23]. The VEGF/VEGFR system plays essential roles in angiogenesis and lymphangiogenesis. Moreover, the levels of VEGF and VEGFR are remarkably elevated in the plasma and the stratum corneum (SC) of lesional skin in AD patients, suggesting that targeting the VEGF/VEGFR system may be a promising treatment for AD [15,16,17,26]. In agreement with findings in AD patients, these phenomena have likewise been observed in animal models of AD [27,28].

Recently, van Aanhold et al. [29] showed that soluble fms-like tyrosine kinase-1 (known as soluble VEGFR-1—a decoy receptor acting as a natural VEGF-A inhibitor) ameliorates skin lesions and inflammation in an AD model of *APOC1* transgenic mice. In addition to neutralizing VEGF-A, inhibitors that can competitively bind to the kinase domains of VEGFR have been reported to also exhibit beneficial effects in in vivo models of inflammatory skin disorders [17,30]. These previous observations are in agreement with our study, in which topical application of Ac-RLYE, a VEGFR-2 antagonist, alleviated inflammatory phenotypes and AD-like symptoms in HDM-challenged mice. As it has been reported that intranasal, but not dermal, administration of HDM in mice [31] increases VEGF and VEGFR-2 expression, resulting in angiogenesis, it is reasonable to suggest that the anti-atopic effect of Ac-RLYE in the current study is probably due to the inhibition of the VEGF/VEGFR-2 system.

We found that Ac-RLYE restored VEGF-induced ZO-1 and VE-cadherin disorganization, which are indicative of TJ and AJ instability, respectively. VEGF is known as a potent inducer of vascular permeability, and disruption of the endothelial barrier function, including barrier instability associated with vascular permeability, is largely regulated by intercellular junctions [32]. Endothelial barrier dysfunction and increased vascular permeability by intercellular junction opening and gap formation between endothelial cells contribute to the extravasation of plasma fluid containing inflammatory cytokines and leukocytes, which is an underlying factor in the pathophysiology of many diseases [33]. These findings could explain how Ac-RLYE relieves AD-like symptoms, as observed in our study, where vascular permeability was restored, and the numbers of infiltrating mast cells into AD lesions were reduced.

The disruption of the epidermal barrier leading to increased permeability of the epidermis has recently been highlighted as a pathophysiological feature of AD [34]. The epidermal differentiation marker filaggrin (FLG) is a key structural protein of the stratum corneum (SC) and a source of natural moisturizing factors, contributing to SC integrity and maintenance of the skin barrier functions [35]. There has been a growing body of evidence that mutations with loss-of-function in *FLG* are strong risk factors for AD and allergic diseases, and *FLG* expression is significantly reduced in AD patients, even without *FLG* mutations [36,37]. In addition, cytokines such as IL-4, IL-13, and IL-17, which are elevated in AD, have been reported to decrease the expression of barrier-related gene product such as filaggrin [38,39]. In the present study, we found that Ac-RLYE significantly suppressed the expression of *IL-4, IL-13,* and *IL-17*. The results suggested that Ac-RLYE restores the skin barrier by regulating the expression of proinflammatory cytokines.

To date, most of the mechanistic studies on atopic dermatitis have demonstrated an imbalance between Th1 and Th2 phenotypes, particularly an enhancement of the Th2 phenotype [40]. Th2 cytokines such as IL-4, IL-5, and IL-13 are assumed to be major players in AD [41]. IL-4, a Th2 cytokine, is essential for IgE synthesis, and elevated IgE production activates immune cells and inflammatory cells, inducing degranulation through intracellular signaling and subsequent hyperactivation [42]. The intensification of the Th1 response with elevated IFN-γ, CXCL9, and TNF-α levels in chronic lesions was also reported [43]. Recently, the expression of Th17 cytokines such as IL-17 and IL-22 in AD lesional skin has been reported [44,45]. IL-17, a hallmark of the Th17 subset, has been reported to regulate Th2 responses, resulting in IL-4 production in a mouse AD model [46]. In this regard, the lack of IL-17A is known to reduce skin inflammation and suppress IL-4 and IgE production. A previous study has revealed that Ac-RLYE possesses anti-inflammatory effects through the inhibition of macrophage infiltration and the polarization of tumor-associated macrophages to the M2 phenotype [11]. In the present study, Ac-RLYE suppressed the mRNA expression of inflammatory cytokines *IL-4, IL-13, IL-31, IL-23,* and *TNF-α* in AD lesions, and this decreased trend of the cytokines was consistent with the decreased production of IgE in serum. Therefore, we conclude that Ac-RLYE can be a promising agent for the treatment for AD through the modulation of immune responses by Th1, Th2, and Th17 activation.

We found that Ac-RLYE treatment noticeably reduced the mRNA expression of *IL-17A*, often known as *IL-17,* which is mainly produced by Th17 cells in AD-like lesions, and the effect was greater than that of Protopic (0.1% tacrolimus ointment) (Figure 6). Although the proportion of Th17 cells in the peripheral blood of AD patients remains controversial, Th17 cell infiltration and IL-17 levels have been reported to increase in eczematous lesions of AD [44]. In the pathogenesis of AD, the increased IL-17 production is known to activate keratinocytes in AD-like lesions and induce the production of exacerbating factors such as cytokines/chemokines (GM-CSF, TNF-α, IL-8, CXCL10) and VEGF [44].

Psoriasis is mediated by Th1 and AD by Th2 cells, but these are two representative inflammatory skin disorders and share general factors exemplified by proinflammatory cytokines, including IL-17 and VEGF [47,48]. We also found that IL-23, a key upstream regulator of IL-17A production that stimulates the differentiation and activation of Th17 cells, was markedly downregulated by Ac-RLYE. Since the IL-23/IL-17 axis is considered crucial in the pathogenesis of psoriasis, and biologics targeting this axis are used to treat psoriasis, the results of the present study provide a bright prospective that Ac-RLYE can also be developed as a treatment for psoriasis.

The skin irritation study of Ac-RYLE in rabbit showed no dermal erythema or edema compared to its negative control. No skin sensitization was also observed in guinea pigs challenged with Ac-RLYE, and therefore this peptide was classified as Grade I (unpublished data). In addition, a 4-week repeated-dose dermal toxicity study followed by a 2-week recovery test in rat showed there were no significant differences in food consumption, body weight gain, relative organ weight gain, and hematological and clinical chemistry parameters between Ac-RLYE-treated and control groups. Although plasma accumulation of Ac-RLYE was observed only in male rats after 4 weeks of repeated dermal administration, it disappeared within 24 h, suggesting that Ac-RLYE does not have any accumulation potential (unpublished data). As it is commonly accepted that the reactivity of human skin is similar to that of guinea pig skin, the above results suggest that Ac-RLYE can be generalized to humans. Based on this compelling evidence, Ac-RLYE is being tested in South Korea in a Phase I clinical study evaluating safety, pharmacokinetics, and efficacy in healthy subjects and patients with AD.

In conclusion, the results of the present study indicate that Ac-RLYE is able to ameliorate AD-like symptoms, in part by suppressing the expression of inflammatory mediators and restoring skin barrier function. These beneficial effects may be associated with the inhibitory effect of Ac-RLYE on vascular hyperpermeability due to elevated VEGF levels in AD-like lesions. Given that all the pre-clinical safety of Ac-RLYE has already been demonstrated, the successful completion of the Phase I clinical trial in Korea will pave the way for the adoption of Ac-RLYE for a new AD therapy.

## 4. Reagents and Materials

### 4.1. Reagents

The peptide Ac-RLYE was obtained from Chempeptide Limited (Shanghai, China) and was at least 98% pure. Cell culture media and supplements were purchased from Invitrogen Life Technologies (Carlsbad, CA, USA). Fetal bovine serum (FBS) was obtained from HyClone Laboratories (Logan, UT, USA), and human basic fibroblast growth factor (bFGF) and VEGF-A were from R&D Systems (Minneapolis, MN, USA). Biostir-AD, an ointment containing the extract of the house dust mite (HDM), *Dermatophagoides farina,* was purchased from Biostir, Inc. (Kobe, Japan). A 0.1% ointment of tacrolimus (Protopic^®^; Astellas Pharma Tech Co., Ltd., Grand Island, NY, USA) was used as a positive control.

### 4.2. Cells

Collagenase type II (Worthington Biochemical Corporation, Lakewood, NJ) was used to isolate human umbilical vein endothelial cells (HUVECs) from umbilical cord veins. The cells were cultured on gelatin-coated plates in Medium 199 supplemented with 20% FBS, 100 U/mL penicillin–streptomycin, 3 ng/mL bFGF, and 5 U/mL heparin. Trypsinization was used to collect the cells (up to passage 7), which were then used for assays.

### 4.3. Immunofluorescence Staining of HUVECs

HUVECs were grown as a monolayer on 2% gelatin-coated cover glasses and cultured in M199 medium containing 1% FBS. The cells were pretreated with 0.15 nM Ac-RLYE for 30 min and stimulated with 10 ng/mL VEGF-A for 1 h. The cells were then fixed with 4% formaldehyde for 15 min at room temperature, permeabilized in 0.1% Triton X-100 for 30 min at 4 °C, and stained overnight at 4 °C with antibodies for rabbit anti-VE-cadherin (1:200, Thermo Fisher Scientific, Waltham, MA, USA) or rabbit anti-zonula occludens-1 (ZO-1, 1:200, Thermo Fisher Scientific, Waltham, MA, USA). Cells were then incubated (1 h) at room temperature with Alexa Fluor 555-conjugated secondary antibodies. To stain the nuclei, the cells were exposed to 4′,6-diamidino-2-phenylindole (DAPI) diluted (1:500) in PBS during the washing step. Dako mounting reagent was used to mount the cells, and a confocal microscope (LSM 700 META, Carl Zeiss, Oberkochen, Germany) was used to examine the results. The levels of VE-cadherin and ZO-1 were quantified using Image J software.

### 4.4. Animals and Ethics Approval

Four-week-old male NC/Nga mice were purchased from Central Laboratory Animal Inc. (Seoul, Korea). Mice were acclimated for one week and maintained at constant temperature (23 ± 2 °C) and humidity (55  ±  15%) under a 12 h light/12 h dark cycle with free access to water and food. All animal experiments were approved by the Animal Care and Use Committee of KNOTUS Co., Ltd. (Incheon, Korea) (protocol number: KNOTUS IACUC 18-KE-290). All in vivo experiments were performed according to the guidelines of the Ethics Committee for Protection and Use of Experimental Animals of KNOTUS Co., Ltd. and in accordance with relevant guidelines and regulations.

### 4.5. Induction of AD-like Skin Lesions and Topical Application

To induce experimental AD-like skin lesions in mouse, we used cutaneous Biostir-AD for challenging NC/Nga mice, according to the manufacturer’s instructions. After 1 week of acclimation, AD was induced. Briefly, the dorsal part of each mouse was shaved with an electric shaver, and then the remaining hair was removed using a depilatory cream. Twice weekly for three weeks, growing hair was shaved with the electric shaver, followed by application of 150 μL of 4% sodium dodecyl sulfate to the treated area (barrier disruption). Biostir-AD (200 mg/mouse) was applied 3 h after barrier disruption. Fifty mice were divided into five groups of five mice per cage (*n* = 10 per group): group 1, control (Biostir-AD + vehicle (distilled water only)); group 2, 0.1% tacrolimus (Biostir-AD + Protopic (positive control)); group 3, 0.004% Ac-RLYE (Biostir-AD + 0.004% Ac-RLYE in distilled water); group 4, 0.02% Ac-RLYE (Biostir-AD + 0.02% Ac-RLYE in distilled water); and group 5, 0.1% Ac-RLYE (Biostir-AD + 0.1% Ac-RLYE in distilled water). Ac-RLYE or 0.1% tacrolimus (100 μL/mouse) was applied daily for 4 weeks. The concentration of Ac-RLYE used in the present study was determined based on the results of previous preliminary experiments (unpublished data). The mice were sacrificed after 4 weeks of Ac-RLYE treatment (Figure 2A, on Day 48).

### 4.6. Measurement of Dermatitis Severity and Ear Thickness

The severity of the dermatitis was measured macroscopically and scored once a week by using a previously established method [49] based on the clinical signs and symptoms of AD, namely, erythema (hemorrhage), edema, excoriation (erosion), and scaling (dryness). The total dermatitis severity score was defined as the sum of the component scores (0, no symptoms; 1, mild; 2, moderate; 3, severe) for each mouse, ranging from 0 to 12.

Ear thickness was measured on the first day of Ac-RLYE application and then once a week using calipers.

### 4.7. Histological Analysis

The dorsal skin tissues were isolated, fixed in 10% formalin immediately after excision from the mice, and embedded in paraffin. Sections of paraffin-embedded samples were stained with hematoxylin and eosin (H&E) for histopathological analysis and with toluidine blue (TB) to visualize and count mast cells. Skin samples were observed and measured with an Olympus BX53 light microscope (Olympus Corporation, Tokyo, Japan). From the histopathological findings, the severity of skin manifestations was assessed on the epidermis (hypertrophy, hyperkeratosis, and infiltration by inflammatory cells) and on the dermis (infiltration by inflammatory cells) and was expressed as the sum of the individual score grades from 0 to 3 (0, no symptoms; 1, mild; 2, moderate; 3, severe), as described previously [50]. The tissue slides were evaluated in a blinded manner by a qualified pathologist.

### 4.8. Analysis of Mouse Blood

Whole blood samples were collected from the mice and used for analysis. For differential cell counts, blood samples were cytospun (1000 rpm, 10 min) (Shandon Cytospin; Thermo Fisher Scientific, Waltham, MA) onto microscope slides and air-dried. The slides were stained with Diff-Quik^®^ (Sysmex Corporation, Kobe, Japan), and differential cell counts were obtained based on morphology and staining characteristics.

### 4.9. Measurement of Skin Barrier Protein Filaggrin (FLG) and Plasma IgE

The dorsal skin was collected from mice after 4 weeks of Ac-RLYE or tacrolimus treatment and stored at −80 °C until use. The dorsal skin tissues were homogenized in RIPA lysis buffer containing a protease inhibitor and a phosphatase inhibitor and centrifuged at 13,000 rpm for 10 min at 4 °C. The supernatant was collected for a protein assay using a Pierce™ BCA Protein Assay Kit (Thermo Fisher Scientific, Inc., Waltham, MA, USA). The level of FLG in mouse skin tissue was quantified using an ELISA kit (MyBioSource, Inc., San Diego, CA, USA) in accordance with the manufacturer’s instructions.

The blood samples were drawn from mice, and the plasma was separated by centrifugation at 10,000× *g* for 10 min at 4 °C and stored at −80 °C until tested. The plasma level of IgE was measured by ELISA according to the manufacturer’s instructions (Mouse IgE ELISA Kit, Abcam, Cambridge, UK).

### 4.10. Real-Time Quantitative RT-PCR

Both the ear and dorsal skin of each mouse were excised using a surgical knife, pooled, and used for RNA isolation. Total RNA was prepared using an RNA isolation kit (GeneAll^®^ Hybrid-R™, GeneAll, Korea) following the manufacturer’s instructions. RNA was quantified using a BioTek Take3^TM^ Multi-Volume plate (Agilent BioTek Instruments, Winooski, VT, USA). An equal amount of total RNA was reverse-transcribed using ReverTra Ace™ qPCR RT Master Mix with gDNA Remover (Toyobo CO., LTD., Osaka, Japan) according to the manufacturer’s instructions. Real time-PCR was performed on a CFX96™ Touch Real-Time PCR System (Bio-Rad, Hercules, CA) using TB Green^®^ Premix Ex *Taq*™ (Takara, Shiga, Japan). Quantification of the target gene expression was determined by the comparative 2^ΔΔ*CT*^ method. The relative expression levels were determined by normalizing expression to glyceraldehyde 3-phosphate dehydrogenase (*GAPDH*). The level of each target gene was expressed as the relative fold change.

### 4.11. Statistical Analysis

Statistical analysis and graphs were prepared using Prism Ver. 5.03 (GraphPad Software Inc., San Diego, CA, USA). Unless otherwise stated, all data were presented as mean ± standard deviation (S.D.), and individual comparisons were made using one-way analysis of variance (ANOVA) followed by Dunnett’s multiple comparison test. For clinical and histopathological score analysis, Kruskal–Wallis’ *H*-test was performed followed by Dunn’s multiple comparison test. Differences were considered statistically significant if *p* < 0.05.

## Figures and Tables

**Figure 1 ijms-23-13498-f001:**
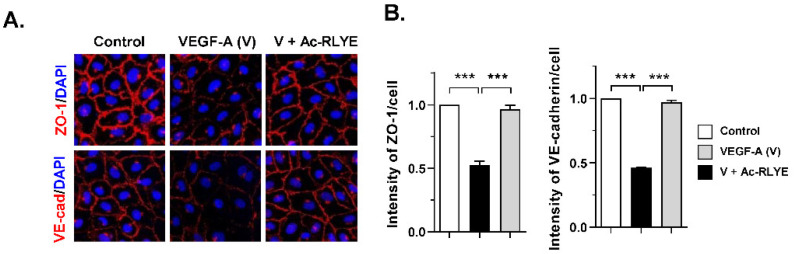
Ac-RLYE inhibits VEGF-A-induced junctional integrity loss in HUVECs. Cells were pre-treated for 30 min with Ac-RLYE (0.15 nM) followed by stimulation with 10 ng/mL VEGF-A (V) for 1 h. (**A**) Confocal images of ZO-1 and VE-cadherin. (**B**) Quantification of ZO-1 and VE-cadherin at contact sites between endothelial cells (*n* = 3). Data are represented as mean ± SD values. *** *p* < 0.001.

**Figure 2 ijms-23-13498-f002:**
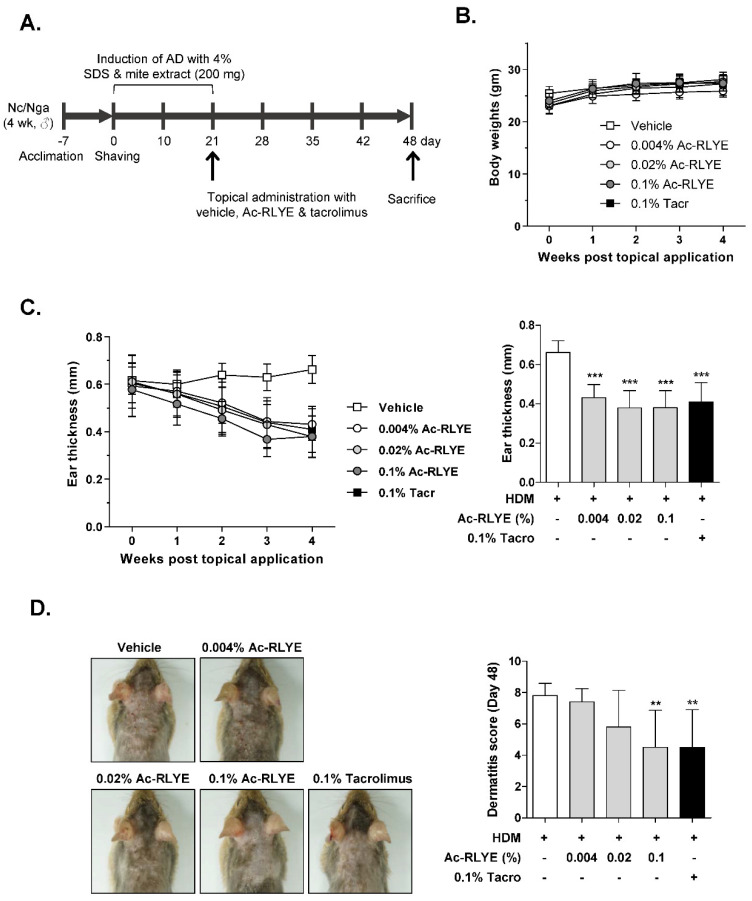
Ac-RLYE alleviates AD-like skin lesions in NC/Nga mice. (**A**) A schematic diagram showing the induction and treatment of AD. (**B**) Body weights were measured for 4 weeks post topical application of Ac-RLYE or tacrolimus (Days 20–48). (**C**) Ear thickness was measured weekly for 4 weeks after AD induction (**left**) and on Day 48 (**right**) (*n* = 10). (**D**) Representative pictures of mouse on Day 48. Dermatitis scores (gross examination) on Day 48 were defined as the sum of scores for four symptoms: erythema (hemorrhage), edema, excoriation (erosion), and scaling (dryness). Data are presented as mean ± SD values for each group (*n* = 10). ** *p* < 0.01, *** *p* < 0.001 in comparison to vehicle-treated group.

**Figure 3 ijms-23-13498-f003:**
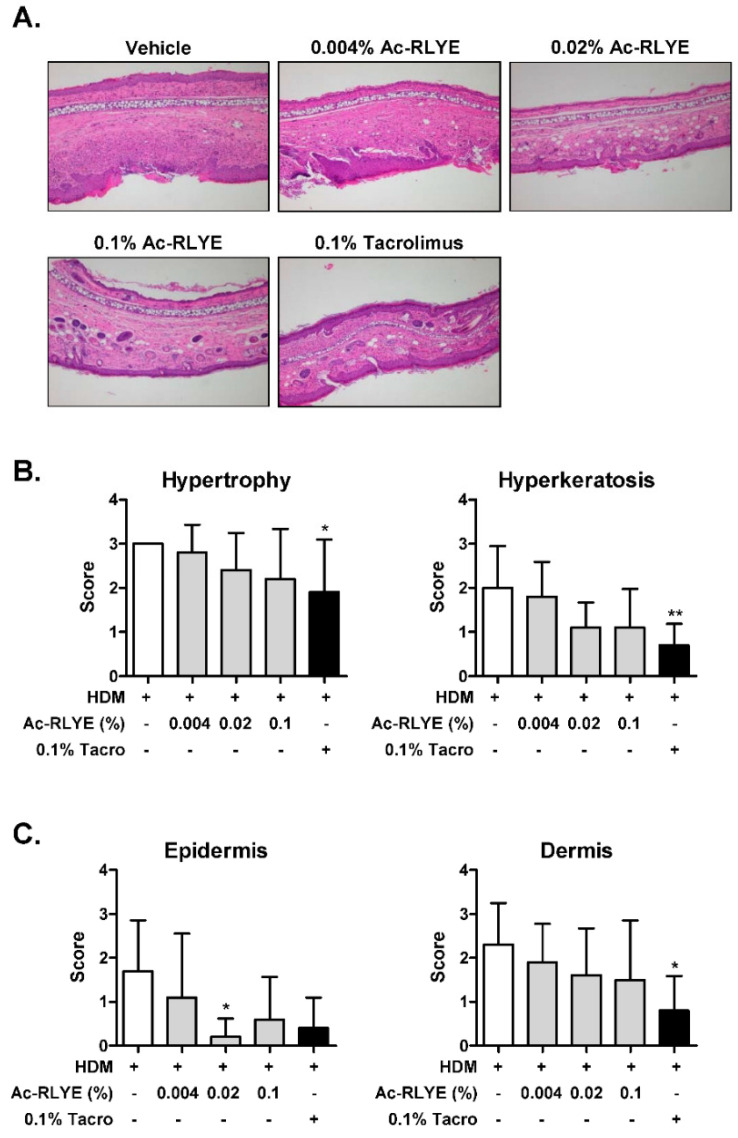
Topical application of Ac-RLYE reduces tissue inflammation in NC/Nga mice. (**A**) Microphotographs of ear sections isolated at Day 48 and stained with H&E. (**B**) The scores of hypertrophy and hyperkeratosis were evaluated. (**C**) The scores of infiltration of inflammatory cells in epidermis and dermis of AD-like skin lesions of NC/Nga mice were evaluated. Data are presented as mean ± SD values for each group (*n* = 10). * *p* < 0.05, ** *p* < 0.01 in comparison to vehicle-treated group.

**Figure 4 ijms-23-13498-f004:**
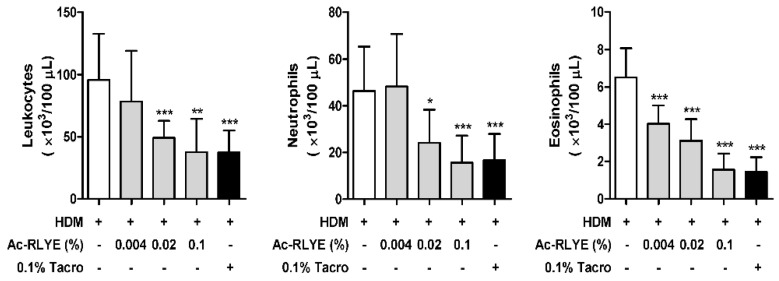
Ac-RLYE reduces immune cell infiltrations. Blood samples were analyzed using differential cell counting. Data are presented as mean ± SD values for each group (*n* = 10). * *p* < 0.05, ** *p* < 0.01, *** *p* < 0.0001 in comparison to vehicle-treated group.

**Figure 5 ijms-23-13498-f005:**
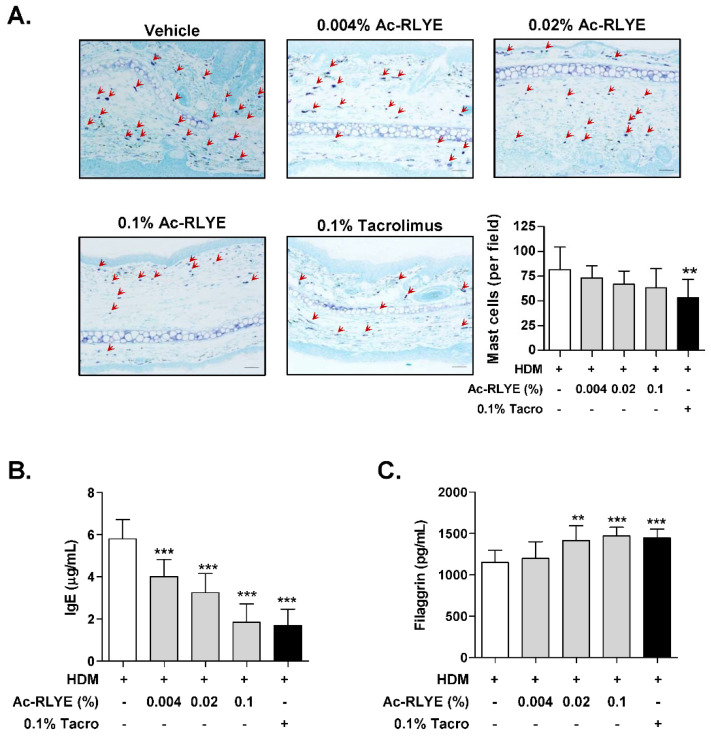
Ac-RLYE inhibits mast cell infiltration and serum IgE and enhances skin barrier repair in AD-like skin lesions of NC/Nga mice. (**A**) Microphotographs of ear sections isolated at Day 48 and stained with toluidine blue. Infiltrating mast cells (red arrows) were counted after toluidine blue staining. Infiltration of mast cells into tissue was observed and counted. (**B**) Serum IgE in mice was measured by ELISA. (**C**) Protein levels of filaggrin in mice skin were evaluated by ELISA. Data are presented as mean ± SD values for each group (*n* = 10). ** *p* < 0.01, *** *p* < 0.001 in comparison to vehicle-treated group.

**Figure 6 ijms-23-13498-f006:**
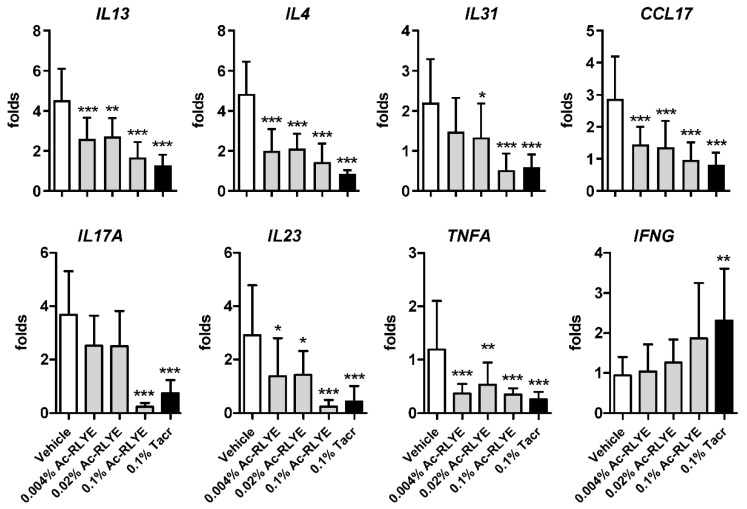
Ac-RLYE decreased the expression of inflammatory cytokines and increased the expression of IFN-γ in AD-like skin lesions of NC/Nga mice. Total RNA was extracted from the dorsal skin of the mice at Day 48. Quantitative PCR was performed to determine the mRNA expression of IL-4, IL-13, TNF-α, CCL17, IFN-γ, IL-17, IL-23, and IL-31. Data are presented as mean ± SD values for each group (*n* = 10). * *p* < 0.05, ** *p* < 0.01, *** *p* < 0.001 in comparison to vehicle-treated group.

## Data Availability

Not applicable.

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
