# Peer review of "Topical Administration of a Novel Acetylated Tetrapeptide Suppresses Vascular Permeability and Immune Responses and Alleviates Atopic Dermatitis in a Murine Model"

_ijms, 2022, doi:10.3390/ijms232113498_

Round 1

Reviewer 1 Report

This paper is generally well-written and reports results suggesting that Ac-RLYE may have beneficial effects in the treatment of atopic dermatitis. However at times the strength of the findings is somewhat overstated, and some additional clarity on the methods used would be useful. Specific comments and suggestions are below.

Abstract, line 19 - 20. Suggest amending the wording for clarity: e.g. '...Moreover, in an in vivo animal model of AD,...'

Section 2.2, lines 98-99. Comments on the methods section are below but the statement that Ac-RLYE was applied once daily for 4 weeks is in conflict with the methods section which states it was applied twice a week. 

Section 2.2, lines 101-102: it is stated that the absence of body weight changes indicate the safety of Ac-RLYE, however this finding is of only limited value in isolation without assessment of other parameters used in toxicity studies. Suggest that direct reference to safety is taken out of this section, instead just noting that no adverse effects on body weight were observed.

Section 2.2, lines 105-107: It is claimed that the effect of Ac-RLYE on ear thickness was greater than that of tacrolimus at the mid and high concentration, however the difference seems marginal at best. 

Section 2.2, lines 121-123: the reduction in dermatitis score was only significantly different from the control at the high concentration of Ac-RLYE, this should be made clearer.

Section 2.2, lines 130-132: It is stated that a dose dependent reduction in hypertrophy and hyperkeratosis, as well as inflammatory cell infiltration, was observed, however, unlike with the positive control, the findings with Ac-RLYE were not significant (with the exception of inflammatory cells in the epidermis only at the mid-dose) and there was a large amount of variability as indicated by the error bars in Figure 3B & 3C. This finding is overstated.

Section 2.4, lines 163-164: again, no significant effect of Ac-RLYE was observed on mast cell infiltration, but this is not made clear in the text. 

Discussion, lines 245-248: it is unclear how the finding that HDM applied intranasally, but not intradermally, increases VEGF/VEGFR2 expression, supports the hypothesis that the effects may be partly due to inhibition of the VEGF/VEGFR2 system, given the HDM was applied dermally in the present study. Perhaps there could be some discussion on how this might be demonstrated in future studies? Why was VEGF/VEGFR2 expression or activation not measured in the present study? 

Discussion, lines 247-249: it is stated that the anti-atopic effect may be partly due to inhibition of VEGF/VEGFR2. It would be useful to have some discussion of whether this mechanism would plausibly explain all the findings, or of what other mechanisms might also be contributing to the effects.

Discussion, lines 273-275: the references cited regarding findings of mechanistic studies to date are quite old (2011 and 2013), are there any more recent ones to support the statement? 

Discussion, lines 308-317: It is useful to know that some toxicity studies have been conducted to support safety, this could perhaps also be mentioned in the introduction as well. In addition to toxicological effects, are there any pharmacological effects from the proposed mode of action that might contribute to side effects in patients?

Discussion lines 319-320: given that some of the effects were only significant at the highest concentration of Ac-RLYE tested, it might be useful to include some discussion about what concentrations of Ac-RLYE might be beneficial clinically, and what is/might be tested in the clinical studies. 

Methods, section 4.3: what method/approach was used to quantify the level of staining in vitro?

Methods, section 4.4: some explanation on how the concentrations tested were selected would be useful. 

Methods, section 4.4: it is unclear if Biostir continued to be applied during the phase when Ac-RLYE or tacrolimus was administered.

Methods, section 4.4: it is not stated where Ac-RLYE or tacrolimius was applied to the mouse (section 2.2 states to the ears, but this should be clearer in the methods section). Note also that 4.4 states mice were treated twice a week whereas 2.2 states treatment was daily. 

Methods, sections 4.7, 4.8, 4.9: these sections could be clearer about the timing of collection of the samples, including whether they were collected at the time the mice were killed.

Methods, section 4.7: should be clear whether the histopathological scoring was blinded to treatment or not.

Methods, section 4.9: states skin was collected 4 weeks post-treatment, does this mean 4 weeks after treatment with Ac-RLYE was completed, or at the end of the 4-week treatment period?

Methods, section 4.10: it is not stated where the RNA was isolated from (Figure 6 states AD-like skin lesions, but this should be clearer in the methods section).

Author Response

Response to Reviewer 1 Comments

Reviewer 2 Report

In this study, the authors tested the efficacy of a novel acetylated tetrapeptide (Ac-RLYE) to treat atopic dermatitis. Ac-RLYE was previously characterized as a selective VEGFR2 signaling inhibitor. Since most AD therapies work by targeting immune response, here the authors propose to use Ac-RLYE to inhibit plasma leakage and movement of inflammatory cells to the skin lesions. While the manuscript is well written and the experiments are well thought out, the figure quality is extremely poor making it hard to understand the results. None of the images and figures are interpretable; all are pixelated. Furthermore, it is not clear if the decreased mast cell infiltration or reduction in inflammatory cytokines are due to reduced plasma leakage (in vivo data for this not shown) and movement of inflammatory cells or because the peptide itself has a direct anti-inflammatory effect.

Author Response

Response to Reviewer 2 Comments

Round 2

Reviewer 2 Report

Authors have addressed all the concerns.
The clarity of the results presented are poor - please correct this before final submission.